# The Interlinking Metabolic Association between Type 2 Diabetes Mellitus and Cancer: Molecular Mechanisms and Therapeutic Insights

**DOI:** 10.3390/diagnostics14192132

**Published:** 2024-09-25

**Authors:** Abutaleb Asiri, Ali Al Qarni, Ahmed Bakillah

**Affiliations:** 1King Abdullah International Medical Research Center (KAIMRC), Eastern Region, Al Ahsa 36428, Saudi Arabia; asiriabu@kaimrc.edu.sa (A.A.); qarniaa@mngha.med.sa (A.A.Q.); 2Division of Medical Research Core-A, King Saud bin Abdulaziz University for Health Sciences (KSAU-HS), Al Ahsa 36428, Saudi Arabia; 3King Abdulaziz Hospital, Ministry of National Guard-Health Affairs (MNG-HA), Al Ahsa 36428, Saudi Arabia

**Keywords:** cancer, diabetes, obesity, inflammation, apoptosis, Akt inhibitors, mTOR inhibitors, PI3K/Akt/mTOR signaling

## Abstract

Type 2 diabetes mellitus (T2DM) and cancer share common risk factors including obesity, inflammation, hyperglycemia, and hyperinsulinemia. High insulin levels activate the PI3K/Akt/mTOR signaling pathway promoting cancer cell growth, survival, proliferation, metastasis, and anti-apoptosis. The inhibition of the PI3K/Akt/mTOR signaling pathway for cancer remains a promising therapy; however, drug resistance poses a major problem in clinical settings resulting in limited efficacy of agents; thus, combination treatments with therapeutic inhibitors may solve the resistance to such agents. Understanding the metabolic link between diabetes and cancer can assist in improving the therapeutic strategies used for the management of cancer patients with diabetes and vice versa. This review provides an overview of shared molecular mechanisms between diabetes and cancer as well as discusses established and emerging therapeutic anti-cancer agents targeting the PI3K/Akt/mTOR pathway in cancer management.

## 1. Introduction

Cancer is a major cause of mortality and a significant health problem worldwide [1]. Type 2 diabetes mellitus (T2DM) is a chronic metabolic disorder characterized by persistent hyperglycemia due to impaired insulin secretion and resistance to peripheral actions of insulin [2,3]. According to the International Diabetes Federation (IDF), approximately 537 million adults (20–79 years) worldwide are living with diabetes in 2021 [4]. T2DM accounts for over 90% of global diabetes [5]. It has been reported that T2DM is associated with an increased risk of various types of cancer such as esophagus, liver, lung, colorectal, and pancreas cancers [5,6,7,8]. Cancer cells accelerate their glucose uptake during tumor progression to support tumor growth [9]. According to epidemiological studies, diabetic individuals are at higher risk of cancer with 8–18% of cancer patients having diabetes [10,11]. The relationship between T2DM and cancer is well established; however, the exact underlying mechanisms are not fully understood. Both T2DM and cancer share common risk factors including obesity and inflammatory processes, hyperglycemia, and hyperinsulinemia [12,13]. These shared risk factors contribute to the development and progression of both diseases, highlighting the importance of maintaining a healthy lifestyle, and early screening for detection and prevention. Additionally, certain signaling pathways associated with both T2DM and cancer including PI3K/AKT/mTOR are used as therapeutic targets for various types of cancers [14,15]. In this review, we summarize the intricate association between T2DM and cancer, exploring the most updated evidence of the underlying mechanisms and targeted therapy drugs used in the management of cancer patients with diabetes.

## 2. Literature Review and Selection

A thorough literature review was conducted to gather relevant studies on the association between cancer and diabetes. Key databases such as PubMed, Google Scholar, and Web of Science were searched using terms related to “cancer”, “diabetes”, “hyperglycemia”, “antidiabetic drugs”, “cancer drug inhibitors”, and “PI3K/AKT/mTOR signaling”. The search strategy involved combining keywords with Boolean operators (AND, OR) to capture a wide range of the relevant literature. Studies were selected based on their relevance to the relationship between diabetes and cancer including observational studies, clinical trials, animal studies, and review articles that addressed epidemiological, clinical, or mechanistic aspects of the link between these two disease conditions. This review focused mainly on type 2 diabetes and its association with different forms of cancer. Key information was extracted from the selected articles, including study objectives, methods, sample sizes, cancer types studied, diabetes types, major findings, and conclusions. Relevant data on clinical trials using specifically PI3K/AKT/mTOR inhibitors were extracted from the ClinicalTrials.com database. This review also included a discussion of potential mechanisms and risk factors linking diabetes and cancer, as reported in the literature.

The inclusion criteria for the articles included the following:Publications that discussed the risk factors for cancer and diabetes.Studies that covered metabolic association between various types of cancer and diabetes.Studies that covered the potential implication of PI3K/AKT/mTOR signaling in the association between cancer and diabetes.Articles published in English within the last two decades, given the rapid advancements.

The exclusion criteria were as follows:Articles not available in full text and studies unrelated to the topic were excluded.Studies that focused solely on type 1 diabetes and its association with cancer.Studies that focused on the use of inhibitors other than Akt inhibitors and mTOR inhibitors.Articles that did not provide substantial information on the association between type 2 diabetes and cancer.Publications in languages other than English.

Overall, the review was structured to provide a comprehensive overview of the current understanding of the metabolic link between cancer and diabetes. The findings were discussed in the context of existing research aiming to present a balanced view of the evidence while acknowledging limitations and future research directions.

## 3. Obesity-Associated Diabetes and Inflammatory Processes in Cancer

Obesity is a well-known risk factor for the development of T2DM and can synergistically increase the risk of cancer [16,17,18]. It has been reported previously that excess body weight is linked to tumorigenesis through various mechanisms involving chronic inflammation, hyperinsulinemia, and anti-hyperglycemic medications [19]. Commonly reported cancers concerning obesity and diabetes include liver, colorectal, pancreas, endometrial, and postmenopausal breast cancers [20]. Moreover, obesity-linked insulin resistance and hyperglycemia increase unbound insulin growth factor-1 (IGF-1) protein in the blood, activating insulin and IGF-1 receptor signaling pathways that ultimately promote tumor growth [21]. Several studies proposed the link between high levels of serum C-peptide, a stable marker of insulin secretion, and high risk of various types of cancer including breast, colorectal, and postmenopausal endometrial cancers [22]. Additionally, obese men with high C-peptide levels are at four times higher risk of mortality from prostate cancer than those with normal C-peptide levels [23]. Furthermore, adipose tissue with impaired adipokine enzymes in obesity-associated T2DM releases factors that promote tumor growth [24]. These factors include adiponectin, leptin, resistin, interleukin-6 (IL-6), phosphoribosyltransferase, and tumor necrosis factor-α (TNF-α) [24]. Obesity-related cancer development occurs due to an imbalance in adipokines, characterized by increased production of leptin (oncogenic adipokine) and reduced release of adiponectin (anti-oncogenic adipokine) [1,25]. Notably, adiponectin, a crucial adipokine produced by fat cells, can activate various signaling pathways including mitogen-activated protein kinase (MAPK), adenosine monophosphate-activated protein kinase (AMPK), and phosphoinositide-3-kinase (PI3K)/protein kinase B (Akt) leading to the suppression of tumor formation induction [26,27]. Additionally, adiponectin triggers tumor suppressor gene liver kinase B1, inhibiting cell invasion, migration, and metastasis, while inducing cytotoxic autophagy [28,29,30]. Moreover, leptin as a key regulator of energy metabolism and the immune system contributes to obesity-linked inflammation as well as influences molecules involved in angiogenesis, proliferation, migration, invasion, and adhesion, particularly in breast carcinogenesis [31]. It has also been found that high leptin levels are associated with an increased risk of colon and breast cancers [32,33,34,35]. The increased expression of leptin receptors has been reported in various cancer types suggesting its impact on cancer progression [36]. Furthermore, it has been reported that cells experience stress due to nutrient excess associated with obesity, leading to reactive oxygen species (ROS) production beyond physiological requirements [37]. The production of ROS may play a role in promoting cancer by enhancing DNA mutation, regulating signaling transduction, and inducing inflammation [37]. Furthermore, obesity-related inflammation enhances the production of various systemic molecules involved in inflammatory responses including interleukin (IL)-1, IL-6, tumor necrosis factor (TNF)-α, and C-reactive protein which may impact tumorigenesis by promoting tumor cell survival and progression [38,39].

## 4. Hyperglycemia and Cancer

Hyperglycemia can lead to the formation of advanced glycated end products (AGEs) and oxidative stress which occur in both T1DM and T2DM [1,10,40]. T2DM is the inability of the body’s cells to absorb and use insulin, leading to the accumulation of glucose in the blood, which is defined as insulin resistance [41,42]. The incidence of hyperglycemia indirectly influences tumor cells by inducing the production of insulin-like growth factor-1 (IGF-1) and inflammatory cytokines in circulation [43]. Furthermore, studies suggest that hyperglycemia has a direct impact on cancer development by promoting cell proliferation, enhancing tumor cell invasion, and inducing resistance to apoptosis [43].

### 4.1. Cancer Cell Proliferation

Increased glucose metabolism in cancer cells is a well-established characteristic known as the Warburg effect (aerobic glycolysis) which leads to enhanced glucose uptake by cancer cells [44,45]. This phenomenon provides high glucose levels to support cancer cells’ rapid proliferation [43]. In vitro studies using cancer cell lines have revealed that elevated levels of glucose lead to increased expression of genes associated with promoting cancer cell proliferation, invasion, and migration [46]. Various proteins have been reported to be implicated in hyperglycemia and cancer cell proliferation [43]. The expression of glucose transporters (GLUTs) including insulin-independent GLUT-1 and GLUT-3 is found to be regulated under hyperglycemic conditions leading to elevation of glucose uptake which therefore promotes cell proliferation [47]. Additionally, high glucose levels induce epidermal growth factor (EGF) expression and epidermal growth factor receptor (EGFR) transactivation, a well-known oncogenic pathway, in pancreatic cancer cell lines which may increase pancreatic cancer cell proliferation [48]. Furthermore, high glucose also induces an aggressive phenotype by stimulating the levels of protein kinase C (PKC) and peroxisome proliferator-activated receptor gamma (PPARγ) [49,50,51]. The overproduction of ROS is a key factor in hyperglycemic complications which can lead to cellular DNA mutations and may play a significant role in cancer initiation and progression [52]. Furthermore, ROS generation is essential for Kras-induced anchorage-independent growth via regulation of the ERK MAPK mitogenic signaling pathway [53]. Moreover, long-term hyperglycemia leads to the production of pro-inflammatory factors such as IL-6, TNF- α, and cyclooxygenase-2 (COX-2) which may play a role in cancer development by stimulating oncogene expression, regulating cell cycle, and promoting cancer cell proliferation [54,55]. Furthermore, hyperglycemia plays a role in promoting breast cancer progression by changing leptin/IGFR1 and Akt/mTOR signaling [56]. Additionally, it contributes to pancreatic cancer progression stimulated by ROS through the suppression of the JNK and c-Jun pathways [57].

### 4.2. Association of Apoptosis and Hyperglycemia in Cancer

Apoptosis is a programmed cell death that is essential to maintain homeostasis [49]. This process is dysregulated in cancer cells leading to uncontrolled cellular growth [44]. In normal cells, elevated levels of glucose trigger apoptosis [58,59]. However, recent studies suggest that cancer cells benefit from glucose metabolism which protects them from cytochrome c-mediated apoptosis [60]. In vitro studies have indicated that the upregulation of transmembrane glucose transporters enhances glucose metabolism in cancer cells [61]. Glucose metabolism is altered in various tumors leading to an increased lactate production, in a transition from aerobic to anaerobic glucose utilization known as the Warburg effect [62,63,64]. Furthermore, cancer cells alter glucose metabolism when exposed to low oxygen levels (hypoxia) due to rapid cell proliferation [65]. Tumors adapt to these changes by stabilizing hypoxia-inducible factor-1α (HIF1α) and translocating it into the nucleus [43]. This process leads to the upregulation of genes associated with metabolism, angiogenesis, cell survival, and anti-apoptosis [66,67]. At normal oxygen levels, HIF1α is degraded by the HIF proly1 hydroxylase (PHD) enzyme to maintain its activity at normal levels [67]. Notably, hyperglycemia interferes with this degradation process leading to an elevated HIF1α level which enhances cancer cell survival and promotes apoptotic resistance [68].

### 4.3. Hyperglycemia and Cancer Metastasis

The spread of cancer cells to other parts of the body is considered a vital step in cancer progression and a significant challenge in cancer treatment, contributing to 90% of cancer-associated deaths [69,70,71,72]. Studies have shown that hyperglycemia plays a role in assisting cancer cell migration and transforming them into secondary sites [40]. Cancer patients with hyperglycemia have a higher rate of metastasis and worse outcomes when compared to non-hyperglycemic patients [43]. Previous reports showed that hyperglycemia led to a 27% increase in lymph node metastasis and a 14.3% increase in liver metastasis and pancreatic cancer, respectively, when compared to patients without hyperglycemia [73]. Furthermore, hyperglycemia decreases the levels of epithelial cell marker E-cadherin and increases the PKC-α pathway leading to a more invasive phenotype [46]. Thus, it is plausible that the epithelial–mesenchymal transition (EMT) is stimulated by high glucose levels resulting in increased cancer cell invasiveness and migration [74]. This process could induce reactive oxygen species (ROS) production which ultimately promotes cancer cell motility and invasiveness [74]. Several studies suggest that hyperglycemia promotes metastasis by EMT induction and vascular destruction through oxidative stress [75]. Additionally, high glucose levels induce multiple signaling pathways known to enhance tumor invasion, such as TGF-β and PI3K/AKT, and increase the migratory effect by impairing the secretion of granulocyte colony-stimulating factor (G-CSF) and inhibiting the mobilization of antitumor neutrophils [76,77].

## 5. Hyperinsulinemia and Cancer

Excessive levels of insulin in the blood as a result of insulin resistance are referred to as hyperinsulinemia. The incidence of insulin resistance leads to impaired consumption of glucose by the cells which therefore results in the accumulation of glucose levels in the blood leading to T2DM [41,42]. Elevating circulating insulin is believed to be a major factor linking diabetes and cancer [78]. Several studies have shown that hyperinsulinemia is a potential risk factor for various types of cancer, including pancreatic, colon, prostate, breast, liver, and kidney cancers, and was associated with a 2-fold risk of cancer mortality [13,79,80,81,82,83,84,85]. The insulin receptor (IR) plays a crucial role in cancer biology [86]. In a hyperinsulinemia state, elevated insulin levels lead to increased production of IGF-1 due to upregulated growth hormone receptor (GHR) signaling [10,87]. The insulin signaling pathway is a crucial intracellular signaling pathway that is responsible for regulating many metabolic processes in the body including growth, survival, and glucose homeostasis [88]. The process is initiated by the binding of insulin and IGF to their respective receptors (IR and IGFR) on the cell surface [88]. Upon binding, the IR is activated and subsequently auto-phosphorylates tyrosine residues which serve as docking sites for downstream effectors including Src homolog 2 (SH2) domains, insulin receptor substrate (IRS), growth factor receptor-bound protein 2 (GRB2), SH2B adaptor protein 2 (SH2B2/APS), and growth factor receptor-bound protein 10 (GRB10) [88]. This binding and subsequent signaling pathway triggered by insulin and its substrates regulates several cellular signaling pathways for metabolism and mutagenesis [89]. Furthermore, the insulin signaling cascade contains key components including RAS, RAF, PI3K, and AKT which are frequently mutated in different types of cancer [90,91]. Thus, alteration in genes involved in the insulin signaling pathway (Figure 1) can influence cancer development. It is still debated whether cancer cells remain insulin-sensitive or acquire insulin resistance to influence cancer progression. In fact, cancer cells can exhibit varying responses to insulin sensitivity and resistance depending on the type of cancer and the specific metabolic adaptation they undergo [13].

### 5.1. PI3K/AKT/mTOR Signaling Pathway in Cancer

The phosphoinositide 3-kinase (PI3K) pathway is important in regulating metabolic homeostasis and insulin sensitivity [92,93]. IRS interaction with the heterodimers of PI3K, a regulatory subunit with an SH2 domain (p85), and a catalytic subunit (p110) stimulates glucose transport within cells [94,95]. p85 plays an effective role in the insulin-mediated PI3K pathway by acting as an adaptor for IRS proteins facilitating the downstream signaling by stabilizing the catalytic activity of the p110 subunit of PI3K [93]. Subsequently, PI3K converts phosphatidylinositol 4,5-bisphosphate (PIP2) to phosphatidylinositol 3,4,5-triphosphate (PIP3) [93,96]. Moreover, high levels of PIP3 are regulated and converted back to PIP2 by phosphatase and a tensin homolog deleted on chromosome 10 (PTEN), a tumor suppress gene frequently mutated in cancer [88,97]. PIP3 triggers the recruitment of proteins containing the plekstrin homolog (PH) domain to the plasma membrane, such as phosphoinositide-dependent kinase 1 (PDK1) and protein kinase B (AKT) [93]. Consequently, PDK1 phosphorylates AKT on the Thr308 residue which further activates the downstream AKT-mediated pathways resulting in phosphorylation and inhibition of Glycogen Synthase Kinase 3β (GSK3β), which in turn activates glycogen synthesis in hepatocytes [93,96,98]. Furthermore, AKT phosphorylates AS160 which plays a key role in the translocation of the glucose transporter GLUT4 to the cell membrane in muscle cells and adipocytes [93,99]. This process increases glucose inflow into the cell. Moreover, AKT directly phosphorylates BAD (BCL-2-associated agonist of cell death) and caspase 9, inhibiting the mitochondrial apoptosis pathway and promoting cell survival [91,100]. Additionally, forkhead family box O (FOXO) phosphorylation and inhibition by AKT can indirectly inhibit apoptosis, because FOXO can promote bcl-2-like protein 11 and induce the expression of the pro-apoptotic cytokine Fas ligand [13,100]. The murine double minute 2 (MDM2) proto-oncogene plays a crucial role in cancer development including promoting cell proliferation, apoptosis evasion, metastasis, and chemotherapy resistance [101,102]. MDM2 is phosphorylated by AKT and can inhibit tumor suppressor TP53, allowing cancer cells to evade cell cycle arrest and apoptosis [13,103]. Additionally, when the mammalian target of rapamycin (mTOR) is phosphorylated by AKT, it activates substrates such as S6 kinase 1 and eIF4E binding protein 1 [104]. These substrates are crucial in regulating mRNA translation initiation and controlling protein synthesis and cell growth [104]. Thus, hyperinsulinemia may contribute to cancer progression by enhancing the translocation of glucose transporters to the cell surface, allowing more glucose to enter the cell, supporting rapid tumor growth.

### 5.2. RAS/MAPK/ERK Signaling Pathway in Cancer

Hyperinsulinemia can also promote cancer progression through the mitogenic pathway known as MAPK/ERK which plays a crucial role in cancer cell proliferation [91,105]. The MAPK/ERK pathway is triggered by the binding between the IR and SHC proteins [88]. This binding plays a key role in recruiting the GRB2-SOS (growth factor receptor-bound 2–son of sevenless) complex [88]. SOS regulates the conversion of GTP to GDP specifically when rat sarcoma (Ras) is associated with the plasma membrane [88,106]. Consequently, SOS-mediated GTP loading activates the Ras protein which is farnesylated and translocated to the cell surface [13]. The Ras protein initiates the cascade signals by activating the Raf protein which subsequently activates mitogen-activated protein kinase (MAPK) [88]. Activated MAPK, also known as an extracellular signal-regulated kinase (ERK1/2), plays a key role in transmitting signals in the mitogenic pathway that promotes cell division, growth, migration, and apoptosis [106,107]. Therefore, hyperinsulinemia enhances farnesylated Ras for GTP loading in response to other growth factors [13]. This effect can promote cancer progression and prevent apoptosis through PI3K/AKT/mTORC and MAPK/ERK signaling pathways [13]. However, the intricate insulin signaling network involves extensive cross-talk between its major components and cross-talk with other signaling networks.

## 6. Insulin-Mediated PI3K Pathway Inhibition in Cancer

For the inhibition of progressive tumors, specific signaling pathways involved in cancer development are targeted to inhibit cell proliferation, regulate the cell cycle, or induce apoptosis [108]. Targeted genes such as PI3K/AKT/mTOR are commonly activated in solid tumors and associated with insulin resistance in diabetes conditions resulting in hyperglycemia [56,109]. The drugs targeting PI3K/AKT/mTOR are shown in Table 1 and Figure 2.

### 6.1. PI3K Inhibitors

PI3K contains several isoforms such as PI3Kα, β, δ, and γ which contribute to different cancer subtypes at various rates [110,111]. Therefore, various inhibitors targeting these isoforms have been synthesized and clinically studied in several clinical trials [112]. Pan-Class I-PI3K inhibitors are a class of targeted agents that were synthesized to inhibit all four PI3K isoforms [113]. This class of medical drugs, such as buparlisib and pictilisib, have shown modest clinical efficacy in clinical trials due to their cellular toxicities which limit the clinical utility of these agents [114,115,116]. Therefore, it is essential to investigate more tolerable treatment options, such as isoform-selective PI3K inhibitors that particularly target specific PI3K isoforms. For instance, BYL719 (Alpelisib) is an oral selective inhibitor that is designed to inhibit PI3Kα (PIK3CA) [116,117]. Preclinical studies revealed that Alpelisib inhibits PI3K signaling, prevents AKT phosphorylation in cell lines with PIK3CA mutations, and blocks tumor growth in xenograft models [118]. The first-in-human phase Ia revealed a tolerable safety profile and promising preliminary activity [119]. This supports the use of Alpelisib as a selective PI3Kα inhibitor in combination with other agents for the treatment of tumors harboring PIK3CA mutations [119]. On the other hand, idelalisib (CAL-101) is an oral-specific inhibitor of the PI3Kδ isoform which has been approved by the Food and Drug Administration (FDA) [120,121]. It has shown an independent inhibition for PI3Kδ isoform activity without affecting PI3K signaling essential for the normal function of healthy cells [122,123]. Idelalisib is the first FDA-approved inhibitor used in combination with rituximab in patients with relapsed or refractory chronic lymphocytic leukemia (CLL) [120,121]. Additionally, it is recommended as a monotherapy for relapsed small lymphocytic lymphoma (SLL) and follicular lymphoma patients who have previously received at least two systemic therapies [124]. Furthermore, Copanlisib is a PI3K α and γ isoform inhibitor that has been approved by the FDA for the treatment of adult patients with relapsed follicular lymphoma (LF) who were previously treated with two or more systemic therapies [125]. This approval provides an additional option to treat patients with this slow-growing type of non-Hodgkin lymphoma by blocking enzymes that promote cell growth [125]. Moreover, Duvelisib is a PI3K δ and γ isoform inhibitor that has also been approved for the treatment of adult patients with CLL and SLL, typically after two or more prior systemic therapies [126]. Despite the clinical development of many PI3K inhibitors, most of these inhibitors were not successfully implemented due to poor clinical response as a result of sub-optimal levels of the drugs when used as monotherapies [111]. Additionally, resistance to these inhibitors can be initiated from parallel signal activation or PI3K pathway reactivation [117]. Inhibition of AKT leads to feedback-mediated regulation of RTK, while the inhibition of mTOR accelerates AKT activity by removing the negative feedback on IRS1 leading to the release of PI3K inhibition through GRB10 [92]. Furthermore, PTEN negatively regulates PI3K signaling, and its loss of expression can lead to resistance to PI3K inhibitors [127]. However, dual inhibitors targeting both PI3K and mTOR can be beneficial to overcome feedback loops and PI3K inhibitor resistance and block PI3K-independent mTOR activation to maximize the efficacy of these inhibitors by suppressing parallel signaling pathways [111,112]. Several PI3K selective and dual PI3K/mTOR inhibitors are being clinically evaluated. BEZ235 (Dactolisib) is a dual inhibitor targeting PI3K and mTOR which is currently undergoing clinical trials [124]. The BEZ235 inhibitor selectively inhibits class I PI3K, mTOR1, and mTOR2 by binding to the ATP-binding kinases and inhibits their catalytic activities and signaling [128]. The BEZ235 dual PI3K/mTOR inhibitor has been reported to exhibit excellent anti-cancer effects in many types of cancer including melanoma, colorectal, lung, renal, breast, and prostate cancer [129,130,131,132,133,134]. Furthermore, PF-04691502 and PF-05212384 (Gedatolisib) are potent ATP-competitive dual class I PI3K/mTOR kinase inhibitors [135,136,137,138]. Preclinical studies showed that blocking PI3K/mTOR with PF-04691502 can enhance TP53/p73 expression and significantly inhibit tumor growth in head and neck squamous cell carcinoma [139]. Moreover, the PF-04691502 inhibitor reduced AKT and S6RP phosphorylation leading to the inhibition of cell proliferation in cancer cells harboring PI3Kα mutations and PTEN deletions [136]. Additionally, preclinical studies showed that PF05212384 inhibitors targeting PI3K and mTOR can reduce AKT/mTOR signaling and induce antitumor activity [137]. It has also been reported that the PF05212384 inhibitor prevents negative feedback loops mediated by mTORC2 resulting in MEK/ERK over-activation in pancreatic cancer [140]. In neuroendocrine tumor cells, treatment with a PF05212384 inhibitor led to cell cycle arrest and induced apoptosis [141]. Therefore, PI3K/mTOR dual inhibitors are promising therapeutic approaches and may exert a more efficient therapeutic effect due to their effective inhibition of the PI3K pathway [124].

Furthermore, blocking the PI3K pathway disrupts insulin signaling which can have a significant impact on insulin resistance [142,143]. PI3K plays a crucial role in the downstream signaling that induces glucose inflow into the cells [144]. The inhibition of the PI3K/AKT signaling pathway could interrupt insulin-mediated glucose uptake and may also lead to systemic hyperglycemia due to the resulting dysregulation in glucose metabolism in multiple tissues including the liver, muscle, and fat [145]. PI3K inhibitors such as pilaralisib and copanisib have shown a 28–30% incidence of hyperglycemia in clinical trials, while other drugs that target the same pathway such as idelalisib did not cause hyperglycemia [142,146]. Understanding the epidemiology and clinical progression of drug-induced diabetes is essential to enhance the efficacy of these inhibitors and manage their clinical complications.

### 6.2. Akt Inhibitors

Akt is essential for various cellular functions including cell growth, survival, proliferation, and glucose metabolism [147,148]. Akt inhibitors are mostly pan-Akt inhibitors due to their three isoforms, Akt-1, 2, and 3 [149]. Genetic alterations in the Akt signaling pathway are frequently observed in many human cancers. Akt overexpression leads to the activation of Akt signaling which was found to be associated with therapy outcomes and resistance to chemotherapy and radiotherapy [150]. Moreover, AKT inhibitors, such as MK-2206 and AZ5363, are currently being evaluated in clinical trials [111,148]. MK-2206 is an allosteric pan-AKT inhibitor that has shown promising in vitro effects, including cell cycle arrest, promotion of apoptosis, and inhibition of cell proliferation in HCC cell lines [151]. Additionally, the MK-2206 inhibitor demonstrates a synergistic activity with cytotoxic compounds such as docetaxel, carboplatin, doxorubin, and gemcitabine in lung cancer [152]. MK-2206 inhibits Akt phosphorylation mediated by gemcitabine and carboplatin, enhancing therapeutic efficacy by suppressing tumor survival [153]. Additionally, recent preclinical studies showed the effectiveness of MK-2206 on acute myeloid leukemia (AML), supporting its use in subsequent clinical trials [154]. MK-2206 also exhibits efficacy in nasopharyngeal carcinoma and PTEN-deficient pancreatic cancer [155]. Furthermore, AZD5363 is a kinase inhibitor that targets all three Akt isoforms [156]. In vitro studies showed that the AZD5363 inhibitor suppresses cancer cell proliferation [148,157]. Preclinical studies suggest that the sensitivity of the AZD5363 inhibitor is associated with the presence of PIK3CA mutations, consistent with other inhibitors in the PI3K/Akt/mTOR signaling pathway [158]. Laboratory investigations have shown that the combination of Akt inhibitors and traditional anti-cancer drugs has improved the therapeutic outcome [111]. However, the complexity of cellular functions of Akt has shifted the interest in Akt inhibitors as monotherapy to combined therapy which has become a major research focus. Furthermore, MK-2206 and AZD5363 inhibitors target AKT which is a key protein in the insulin signaling pathway that can lead to insulin resistance [152,156]. Studies have reported that MK-2206 inhibitors can inhibit insulin-mediated glucose metabolism and protein synthesis in skeletal muscles by blocking the phosphorylation of AKT and its downstream targets [159]. Moreover, the inhibition of AKT by AZD5363 has been found to cause a reversible increase in blood glucose levels indicating its impact on insulin sensitivity [156]. This suggests that MK-2206 and AZD5363 inhibitors can induce insulin resistance by interfering with the insulin signaling pathway that regulates glucose metabolism.

### 6.3. mTOR Inhibitors

Rapamycin and other rapalogs (everolimus, temsirolimus, and deforolimus) are the first generation of mTOR inhibitors which act as allosteric inhibitors targeting mTOR complex 1 (mTORC1) [160,161]. While these rapalogs have been approved for treating various cancers, rapamycin is primarily used as an immunosuppressant to prevent transplant rejection [111,162,163]. Rapamycin inhibits the phosphorylation of the mTORC1 protein kinase S6K1 by binding with the protein FKBP-12 to form a complex that can bind with the FKB region of mTOR resulting in the inhibition of mTORC1 activity [164]. This inhibition leads to the suppression of protein synthesis, and cancer cell growth, and induces autophagy in tumor cells [164]. Second-generation inhibitors of mTOR targeting mTORC1 and mTORC2 such as BI860585, DS-3078a, GDC-0349, Sapanisertib, and CC-223 are currently undergoing clinical trials [165,166,167,168,169]. ATP-competitive mTOR inhibitors are small molecule analogs that compete with ATP to occupy mTOR kinase active sites and provide full blockade of mTORC1 and mTORC2 [161]. This blockade inhibits Akt phosphorylation caused by mTORC2 and observed resistance to rapamycin analogs [161]. For instance, the AZD8055 inhibitor prevents Akt phosphorylation from subsequently inducing autophagy in cancer cells and inhibits tumor growth in vivo [170]. However, mutations in the mTOR gene, as well as loss of function mutations in TSC1, TSC2 (tuberous sclerosis proteins 1 and 2), or STK11 (serine/threonine kinase 11), make cancer cells sensitive to mTOR inhibitors [171,172]. Nevertheless, mutations in the FKBP domain of mTOR lead to resistance to allosteric mTOR inhibitors [173]. Although these mutations in the FKBP domain are relatively uncommon in most cancer subtypes, they remain sensitive to ATP-competitive mTOR kinase inhibitors [174]. Preclinical studies have shown that mTOR kinase inhibitors are more effective against both acquired and intrinsic rapamycin-resistant cancer cells [175]. Studies indicate that downstream signaling pathways like RAF/MEK/ERK may compensate for the loss of mTOR activity, highlighting the need for combination therapies [176,177]. Thus, comprehending these resistance mechanisms and the sensitivity to existing PI3K/AKT/mTOR pathway inhibitors is essential to enhance the effectiveness of these inhibitors and patient selection as well as improve the success of combined therapies. Additionally, mTOR inhibitors are associated with a high incidence of hyperglycemia and new-onset diabetes with 13–50% of patients treated with these inhibitors [178]. The mechanisms leading to hyperglycemia are believed to be caused by impaired insulin secretion and insulin resistance [178]. In vivo studies have shown that treatment with rapamycin decreased insulin sensitivity in parallel with AKT phosphorylation and enhanced GSK3β activity leading to a reduction in glycogen synthesis [179,180]. It has also been reported that the rapamycin inhibitor increases JNK activation, suggesting that rapamycin might induce insulin resistance via the JNK pathway [149]. Moreover, the selective mTOR kinase inhibitor AZD8055 has been associated with hyperglycemia and insulin resistance in animal studies [170,180]. It was found that AZD8055 decreases insulin-stimulated glucose uptake by 30% in muscle tissues without affecting GLUT4 translocation [170]. Furthermore, hyperglycemia caused by mTOR inhibitors can also result from reduced insulin secretion [178]. Since mTORC1 has been shown to improve insulin secretion, it is plausible that mTOR inhibition may decrease insulin secretion by β-cells [181,182]. Further studies showed that mTOR inhibition by rapamycin resulted in decreased glucose-induced insulin secretion by 33% in β-cells [179]. It was reported that treatment with rapamycin significantly reduced β-cell mass by 50% in diabetic animals as a result of increased β-cell apoptosis induced by JNK activation in islets by mTOR inhibition [179]. These findings indicate that mTOR inhibitors can disrupt insulin production by β-cells resulting in decreased glucose-induced insulin secretion which may ultimately cause hyperglycemia.

## 7. Antidiabetic Drugs and Their Anti-Cancer Effects

Antidiabetic drugs can reduce cancer risk by directly affecting the metabolism in cancer cells and indirectly by impacting risk factors associated with malignancy [183]. Numerous clinical studies have shown that antidiabetic drugs are linked to a significant reduction in cancer incidence and mortality [10,184,185]. Although there are many antidiabetic drugs that are currently available, certain medications such as biguanides (metformin), thiazolidinediones (TZDs), and sulfonylureas (SUs) have shown anti-proliferative effects on various cancer cell types [186,187]. Metformin is the most commonly used biguanide drug prescribed for T2DM [188,189]. Several studies have reported that metformin has reduced cancer progression and mortality in T2DM patients [190,191,192,193]. Furthermore, metformin helps in lowering the risk of cancer incidence in patients treated with SUs or insulin [194]. The potential mechanism for metformin’s anti-cancer properties may include the inhibition of mTOR signaling by activating the adenosine monophosphate (AMP)-activated protein kinase (AMPK)-dependent pathway which disturbs protein synthesis and suppresses cell growth and proliferation by reducing insulin levels and glucose uptake [195,196,197]. AMPK activation may also lead to p53 phosphorylation which prevents cell invasion and metastasis [198]. Additionally, metformin can exert anti-cancer properties through the AMPK-independent pathway by inducing autophagy and apoptosis [199]. It also reduces ROS production by suppressing mitochondria complex 1 resulting in a reduction in DNA damage which inhibits cancer development [200]. TZDs (pioglitazone, rosiglitazone, and troglitazone) are approved drugs for T2DM patients and were found to exhibit [201] anti-cancer properties in preclinical and clinical studies [202,203]. TZDs promote insulin sensitivity by activating peroxisome proliferator-activated receptor gamma (PPAR-γ) which has been found to suppress tumor growth in various types of tumor cells by the downregulation of proliferation and insulin resistance and induction of apoptosis [1,204,205]. An in vitro study found that the PPAR-γ activation leads to the inhibition of the PI3K/AKT pathway which induces cell cycle G2 arrest and inhibits cell proliferation in bladder cancer [206]. Furthermore, pioglitazones have shown antitumor effects through the inhibition of the MAPK/AKT cascade resulting in a reduction in cell proliferation and invasiveness as well as an induction of apoptosis in NSCLC cells [207]. Glibenclamide is a sulfonylurea drug for T2DM patients that targets sulfonylurea receptors (SURs) in pancreatic β-cells [208,209]. SURs are subunits of adenosine triphosphate-sensitive potassium channels (KATP channels) [209]. Upon glibenclamide binding, it blocks these receptors leading to cell depolarization which subsequently opens voltage-gated calcium channels [210,211]. This allows calcium to enter the cell, and ultimately, insulin secretion occurs through exocytosis [211,212]. Several studies have suggested that glibenclamide may inhibit tumor growth in different types of cancer, but the exact mechanism remains unclear [213,214,215]. SUR belongs to the adenosine triphosphate-binding cassette (ABC) protein superfamily which utilizes ATP to transport various substances across cell membranes such as metabolic products, lipids, and drugs [209,216]. Multidrug-resistant proteins (MRPs) are a subgroup of ABC transporters found overexpressed in tumor cells leading to chemotherapy resistance, making it a potential target for anti-cancer therapies [209,217,218]. Glibenclamide has been found to cause accumulation of calcein, an MRP1 substrate, with overexpressed MRP1 in lung cancer cells, indicating its inhibitory role for MRP1 [218,219]. It was reported that glibenclamide suppresses cell growth, cell cycle progression, EMT, and cell migration in NSLC expressing SUR [220]. Moreover, it was also reported that glibenclamide exerts anti-cancer properties in gastric cancer cells expressing the KATP ion channel through the induction of ROS production and apoptosis of the cells by activating a pro-apoptotic c-Jun N-terminal kinase and inhibiting the anti-apoptotic kinase AKT kinase activity [218,221]. These findings highlight the potential of glibenclamide in cancer therapy. Furthermore, recent studies showed that sodium/glucose transporter 2 (SGLT2) inhibition through uridine medication can reduce the risk of prostate cancer [222]. Additionally, glucagon-like peptide 1 receptor agonists (GLP-1RAs), relatively new injectable antidiabetes agents, were found to be associated with a reduction in colorectal cancer in drug-naive patients with T2DM [223].

## 8. Conclusions

T2DM and cancer are complex diseases that share common mechanisms including obesity, inflammatory stress, chronic hyperglycemia, and insulin resistance. These mechanisms are involved in the association with tumor development and progression by promoting cell growth, proliferation, migration, and invasion as well as inhibiting apoptosis in tumor cells. Understanding these mechanisms linking both diseases is crucial for developing effective methods for diagnosis, prevention, and treatment. Furthermore, certain pathways such as the PI3K/AKT/mTOR pathway, which are associated with glucose metabolism, frequently exhibit molecular alterations in various cancer subtypes. To overcome the impact of drug resistance and toxicities, a therapeutic combination targeting these specific pathways could provide a promising intervention strategy to reduce the burden impact in both diseases. However, the inhibition of PI3K/AKT/mTOR by small molecule inhibitors may lead to insulin resistance in cancer cells, posing a significant challenge that could worsen disease outcomes. Identifying predictive biomarkers for effective treatments involving PI3K/AKT/mTOR inhibitors is essential to anticipate potential complications. Moreover, antidiabetic drugs appear to reduce cancer risk. Understanding the precise mechanism by which these treatments prevent cancer may help identify novel strategies to treat cancer patients and prevent the disease’s incidence. Further research into the cellular and molecular interactions driving the intricate relationship between T2DM and cancer is essential. This will enhance our understanding and improve the clinical outcomes of patients affected by both diseases.

## Figures and Tables

**Figure 1 diagnostics-14-02132-f001:**
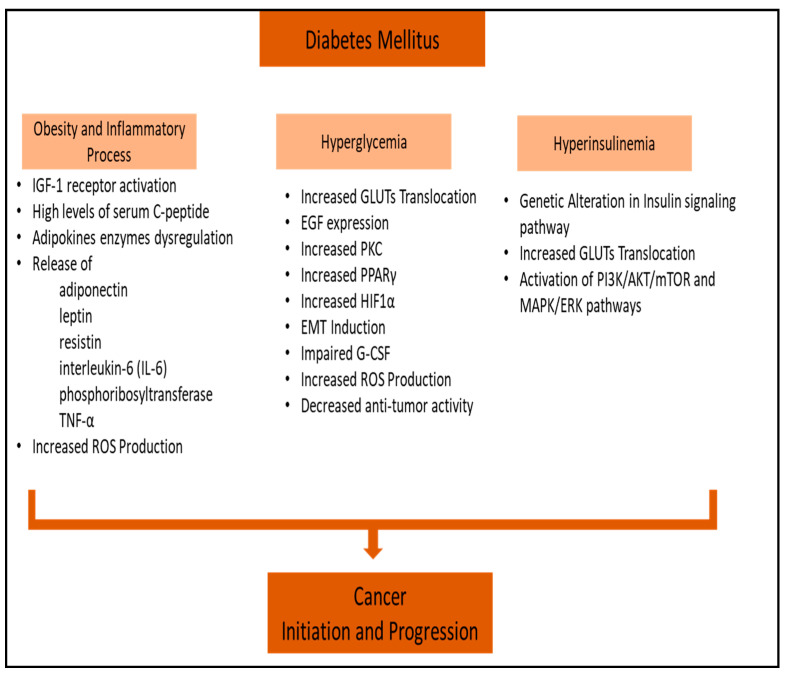
Schematic overview of mechanisms underlying the association between diabetes mellitus and cancer. Akt: protein kinase B, EGF: epidermal growth factor, EMT: epithelial–mesenchymal transition, ERK: extracellular signal-regulated kinase, GLUTs: glucose transporters, G-CSF: granulocyte colony-stimulating factor, HIF1α: hypoxia-inducible factor-1α, IGF-1: insulin growth factor-1, MAPK: mitogen-activated protein kinase, PPARγ: peroxisome proliferator-activated receptor gamma, PI3K: phosphoinositide-3-kinase, mTOR: mammalian target of rapamycin, PKC: protein kinase C, ROS: reactive oxygen species, and TNF-α: tumor necrosis factor-α.

**Figure 2 diagnostics-14-02132-f002:**
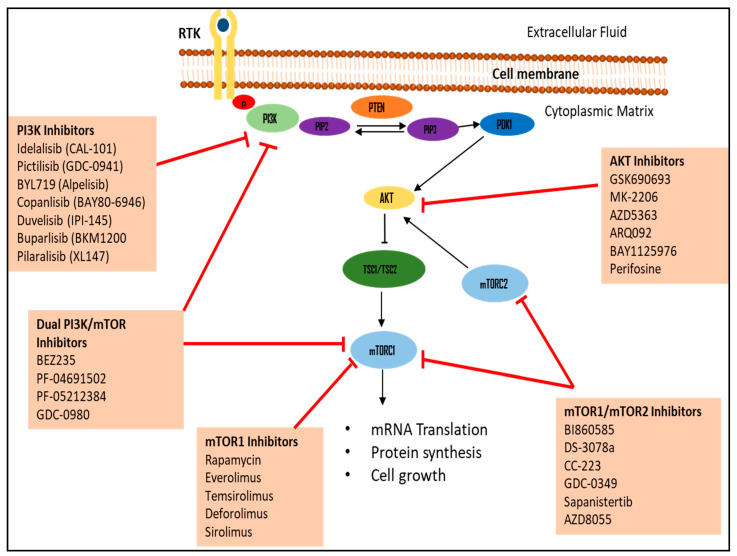
The PI3K/AKT/mTOR pathway and associated inhibitors. The PI3K/AKT/mTOR is a major intracellular pathway that promotes cell growth, proliferation, and migration and inhibits apoptosis in tumor cells. Upon the activation by RTK binding, PIP2 is converted to PIP3 via phosphorylation by PI3K. This process is negatively regulated by PTEN. AKT and PDK1 are then recruited to the plasma membrane which leads to the phosphorylation of AKT by PDK1. This phosphorylation of AKT subsequently inactivates TSC2 leading to the degradation of TSC1/TSC2 and activation of mTOR leading to mRNA translation, protein synthesis, and cell growth. The activation of these molecules promotes tumorigenesis. PI3K, AKT, and mTOR are therapeutic targets that act as inhibitors. Akt: protein kinase B, mTORC1: mammalian target of rapamycin complex 1, mTORC2: mammalian target of rapamycin complex 2, PTEN: phosphatase and tensin homolog deleted on chromosome 10, PIP3: phosphatidylinositol 3,4,5-triphosphate, PIP2: phosphatidylinositol 4,5-bisphosphate, PI3K: phosphoinositide-3-kinase, PDK1: phosphoinositide-dependent kinase 1, RTK: receptor tyrosine kinase, TSC1: tuberous sclerosis proteins 1, and TSC2: tuberous sclerosis proteins 2.

**Table 1 diagnostics-14-02132-t001:** PI3K/AKT/mTOR inhibitors in clinical trials.

Target	Inhibitor	Disease Condition	Sample Size	Clinical Phase	Clinical Trial ID *
PI3K	Idelalisib (CAL-101)	Non-Hodgkin’s Lymphoma	125	II	NCT01282424
PI3K	Pictilisib (GDC-0941)	Non-Hodgkin’s Lymphoma	60	I	NCT00876122
PI3K	BYL719 (Alpelisib)	Advanced Solid Malignancies	221	I	NCT01219699
PI3K	Copanlisib (BAY80-6946)	Advanced Cancer	57	I	NCT00962611
PI3K	Duvelisib (IPI-145)	Advanced Hematologic Malignancies	210	I	NCT01476657
PI3K	Buparlisib (BKM120)	Relapsed or Refractory Non-Hodgkin Lymphoma	7	I	NCT01719250
PI3K	Pilaralisib (XL147)	Endometrial Carcinoma	67	II	NCT01013324
Dual PI3K/mTOR	BEZ235	Advanced Solid Tumors	33	I	NCT01343498
Dual PI3K/mTOR	PF-04691502 PF-05212384	Recurrent Endometrial Cancer	67	II	NCT01420081
Dual PI3K/mTOR	GDC-0980	Refractory Solid Tumors or Non-Hodgkin’s Lymphoma	121	I	NCT00854152
mTOR	AZD8055	Gliomas	22	I	NCT01316809
mTOR	Temsirolimus	Breast Neoplasms	108	II	NCT00062751
mTOR	Sirolimus	Advanced Cancers	40	I	NCT00707135
mTOR	BI860585	Advanced and/or Metastatic Solid Tumors	90	I	NCT01938846
mTOR	DS-3078a	Advanced Solid Tumors or Lymphomas	32	I	NCT01588678
mTOR	GDC-0349	Advanced or Metastatic Solid Tumors or Non-Hodgkin’s Lymphoma	10	I	NCT01356173
mTOR	Sapanisertib (MLN0128)	Advanced Malignancies	198	I	NCT01058707
mTOR	CC-223	Advanced Solid Tumors, Non-Hodgkin Lymphoma, or Multiple Myeloma	226	I/II	NCT01177397
mTOR	Everolimus	Advanced Solid Tumors	30	II	NCT02201212
mTOR	Ridaforolimus	Refractory or Advanced Malignancies	147	I	NCT00112372
mTOR	Deforolimus (AP23573)	Relapsed or Refractory Hematologic Malignancies	57	II	NCT00086125
AKT	GSK690693	Solid Tumors and Lymphoma	70	I	NCT00493818
AKT	MK-2206	Metastatic Neuroendocrine Tumors	11	II	NCT01169649
AKT	AZD5363	Advanced Solid Tumors	12	I	NCT03310541
AKT	ARQ092	Advanced Solid Tumors and Recurrent Malignant Lymphoma	120	I	NCT01473095
AKT	BAY1125976	Neoplasm	79	I	NCT01915576
AKT	Perifosine	Recurrent/Progressive Malignant Gliomas	32	II	NCT00590954

* Data obtained from ClinicalTrials.gov database.

## Data Availability

All data used are drawn from the literature and available in PubMed and/or ClinicalTrials.gov.

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
