# Peer review of "The Interlinking Metabolic Association between Type 2 Diabetes Mellitus and Cancer: Molecular Mechanisms and Therapeutic Insights"

_diagnostics, 2024, doi:10.3390/diagnostics14192132_

Round 1

Reviewer 1 Report

Comments and Suggestions for Authors

Hello Authors, I found your review article to be very insightful, particularly regarding the molecular markers and risk factors that connect Type 2 Diabetes Mellitus (T2DM) and Cancer, with a focus on the PI3K/Akt/mTOR signaling pathway. I believe that your article is of high quality and should be published. I recommend submitting it for publication. Additionally, I suggest having it edited by professional English writing services. Congratulations on your outstanding work!

Comments on the Quality of English Language

There are no comments on the quality of the English, and if possible, I would recommend professional English writing services edit it.

Author Response

Reviewer #1:

Comments and Suggestions for Authors:

Hello Authors, I found your review article to be very insightful, particularly regarding the molecular markers and risk factors that connect Type 2 Diabetes Mellitus (T2DM) and Cancer, with a focus on the PI3K/Akt/mTOR signaling pathway. I believe that your article is of high quality and should be published. I recommend submitting it for publication. Additionally, I suggest having it edited by professional English writing services. Congratulations on your outstanding work!

Comments on the Quality of English Language:

There are no comments on the quality of the English, and if possible, I would recommend professional English writing services edit it.

Authors Response:

We are pleased to hear that you find our review article very insightful and of high quality.

Thank you!

As you suggested, we have proof edited the manuscript and corrected grammatical mistakes and misspelling.

Reviewer 2 Report

Comments and Suggestions for Authors

The present review is aimed to highlight interconnections between T2DM and cancer at the molecular level, as well as to discuss therapeutic approaches that could benefit patients with association of these pathologies.

The topic of the review is very important due to the high social impact of the abovementioned pathologies.

I have a few suggestions to improve the manuscript though:

1.    The review needs to be more structured. There is a section 4.2, for example, but no section 4.1. Please, correct. Some paragraphs are too big and occupy the full page – they should be divided into the smaller ones. Section 3.2 needs to be renamed: Association of apoptosis and hyperglycemia in cancer, for example. As hyperglycemia is still in focus here, not apoptosis.

2.    Lines 193-222 describe the normal insulin signaling. Authors state that hyperinsulinemia may contribute to cancer through the increased proliferation and inhibition of apoptosis. But this contradicts the statement above that “alteration of genes involved in the insulin signaling pathway can influence cancer development”, as normal insulin signaling will not lead to cancer. It is also worthy to describe here if cancer cells remain insulin sensitive or acquire insulin resistance, and if it influences cancer progression.

3.    Abbreviations used in figures have to be deciphered.

4.    Sentence in lines 303-305 duplicates the meaning of sentence 301-303.

5.    It is worthy to mention insulin resistance not just hyperinsulinemia in conclusions.

Author Response

Reviewer #2:

The present review is aimed to highlight interconnections between T2DM and cancer at the molecular level, as well as to discuss therapeutic approaches that could benefit patients with association of these pathologies. The topic of the review is very important due to the high social impact of the abovementioned pathologies.

I have a few suggestions to improve the manuscript though:

  1. The review needs to be more structured. There is a section 4.2, for example, but no section 4.1. Please, correct. Some paragraphs are too big and occupy the full page – they should be divided into the smaller ones. Section 3.2 needs to be renamed: Association of apoptosis and hyperglycemia in cancer, for example. As hyperglycemia is still in focus here, not apoptosis.
  2. Lines 193-222 describe the normal insulin signaling. Authors state that hyperinsulinemia may contribute to cancer through the increased proliferation and inhibition of apoptosis. But this contradicts the statement above that “alteration of genes involved in the insulin signaling pathway can influence cancer development”, as normal insulin signaling will not lead to cancer. It is also worthy to describe here if cancer cells remain insulin sensitive or acquire insulin resistance, and if it influences cancer progression.
  3. Abbreviations used in figures have to be deciphered.
  4. Sentence in lines 303-305 duplicates the meaning of sentence 301-303.
  5. It is worthy to mention insulin resistance not just hyperinsulinemia in conclusions.

Authors Responses:

We are very thankful for the reviewer’s overall positive comments on the manuscript and the suggestions to help us improve the manuscript.

R1: There was a section 4.1 “PI3K/AKT/mTOR Signaling Pathway in Cancer” just below Fig.1 legend. As suggested, we have renamed Section 3.2 to be “Association of Apoptosis and Hyperglycemia in Cancer”. Thank you for the suggestion.

R2: We apologize for the lack of clarity. We have now rephrased the sentence to make it clearer (lines 326-331). We have also included the following sentence as suggested “xxx”. Thank you for pointing out this important observation.

R3: All the abbreviations in the figures were defined and included in the corresponding figure legend.

R4: We have omitted the duplicated sentence. Thank you for highlighting this point.

R5: We have included insulin resistance in the conclusion (lines 511-513). Thank you for this suggestion.
